# 3D Structure Prediction of Atomic Systems with Flow-Based Direct Preference Optimization

**Rui Jiao**[1,2]  **Xiangzhe Kong**[1,2]  **Wenbing Huang**[3,4*]  **Yang Liu**[1,2*]

[1]Dept. of Comp. Sci. & Tech., Institute for AI, Tsinghua University
[2]Institute for AIR, Tsinghua University
[3]Gaoling School of Artificial Intelligence, Renmin University of China
[4] Beijing Key Laboratory of Big Data Management and Analysis Methods, Beijing, China

## Abstract

Predicting high-fidelity 3D structures of atomic systems is a fundamental yet challenging problem in scientific domains. While recent work demonstrates the advantage of generative models in this realm, the exploration of different probability paths are still insufficient, and hallucinations during sampling are persistently occurring. To address these pitfalls, we introduce FlowDPO, a novel framework that explores various probability paths with flow matching models and further suppresses hallucinations using Direct Preference Optimization (DPO) for structure generation. Our approach begins with a pre-trained flow matching model to generate multiple candidate structures for each training sample. These structures are then evaluated and ranked based on their distance to the ground truth, resulting in an automatic preference dataset. Using this dataset, we apply DPO to optimize the original model, improving its performance in generating structures closely aligned with the desired reference distribution. As confirmed by our theoretical analysis, such paradigm and objective function are compatible with arbitrary Gaussian paths, exhibiting favorable universality. Extensive experimental results on antibodies and crystals demonstrate substantial benefits of our FlowDPO, highlighting its potential to advance the field of 3D structure prediction with generative models.

## 1 Introduction

Predicting 3D structures of atomic systems is indispensable in various scientific domains, ranging from pharmaceutical drug design [1, 17] to materials science [7]. Accurate 3D modeling is not only crucial for understanding the physical and chemical properties of substances at the atomic level [2, 18] but also for simulating and predicting their behavior in various environments [3, 26]. Nevertheless, it remains challenging due to the intricate nature of atomic interactions, the vastness of the conformational space, as well as limited resources of structure data.

Conventional methods typically employ physics-based algorithms to derive structures at local energy optimum [24, 31, 32]. Recent advancements leverage deep generative model to learn the distribution of stable structures from available data, showcasing remarkable success across various domains. For example, DiffAb [20] designs a diffusion-based method for antigen-specific antibody design, which is further available for antibody structure prediction, and DiffCSP [14] proposes a joint diffusion framework for crystal structure prediction. Despite these advancements, generative models for structure prediction are confronted with two primary challenges.

First, existing structure prediction methods predominantly utilize diffusion-based generative models. While effective, this focus narrows the scope of exploration into other probability paths that could

---

*Wenbing Huang and Yang Liu are corresponding authors.

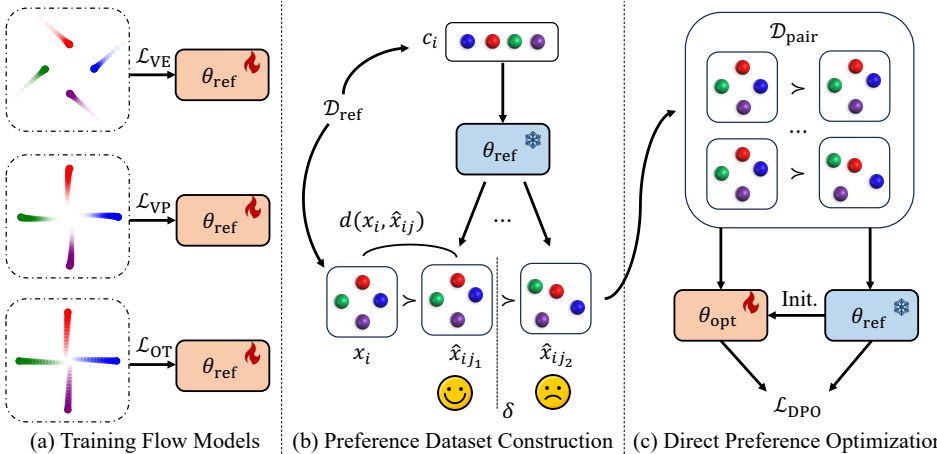

| (a) Training Flow Models | (b) Preference Dataset Construction | (c) Direct Preference Optimization |

Figure 1: Overview of the proposed FlowDPO pipeline. As described in Section 3.1, the process begins by training a flow matching model, denoted as $\theta_{\text{ref}}$, using an arbitrary pre-defined Gaussian path. Next, as outlined in Section 3.2, we construct a preference dataset, $\mathcal{D}_{\text{pair}}$, by evaluating the distances between generated samples $\hat{x}_{ij}$ and the ground structure $x_i$ under a given context condition $c_i$—such as an antibody sequence or crystal composition. These samples are derived from the reference training set $\mathcal{D}_{\text{ref}}$. This dataset is then used to fine-tune the model $\theta_{\text{opt}}$ through the DPO training objective $\mathcal{L}_{\text{DPO}}$, detailed in Section 3.3.

potentially offer substantial benefits. A notable example is the Optimal Transport (OT) path, which has recently been demonstrated to be particularly effective in the field of molecular generation [28]. Second, current training paradigm frequently leads to hallucinated distribution peaks [1]. Most generative models are trained through maximizing the likelihood or its lower bound on the ground-truth structures, which are easily haunted by hallucinations due to the lack of negative samples during training. In the field of natural language processing or computer vision, Direct Preference Optimization (DPO) [25, 30] is proposed to align the model with human preferences, which effectively reduces hallucinations. For 3D structure prediction, such preferences can be naturally extended to similarity with the reference structure (*e.g.* RMSD). However, it remains unclear whether the DPO method is compatible with arbitrary probability paths.

To address the above pitfalls, we introduce FlowDPO, a novel framework that explores flexible selection of Gaussian paths and enhances the quality of generated structures by alignment with the reference distribution. Specifically, we approach the structure prediction task via flow matching models regarding various paths. Given a pre-trained flow matching model, we sample multiple structures for each entry in the training set, evaluate these candidates against known ground truths to compute similarity, and construct an automatic preference dataset. Notably, we theoretically derive the unified objective of DPO for arbitrary Gaussian paths, and leverage the preference to enhance the performance of the original generative model. Intuitively, such a paradigm not only augment data with self-distilled samples, but also endow the model with the ability to distinguish between high-fidelity and hallucinated samples.

In summary, our contributions are threefold:

- We explore multiple accessible probability paths for the 3D structure prediction task, and to the best of our knowledge, we are the first to theoretically prove the compatibility of DPO with arbitrary Gaussian paths by deriving a unified objective.

- Based on the theoretical results, we develop a novel framework to encourage better alignment of flow matching models with desired reference distribution in 3D structure prediction, which effectively suppresses the probability of hallucinations.

- Our approach yields promising results on antibody and crystal structure prediction tasks, showcasing the versatility and efficacy of our FlowDPO.

## 2 Related Work

**Structure Prediction for Atomic Systems.** 3D Structure prediction, including predicting conformations from molecular topological graphs [34], determining unit cell structures from crystal compositions [14], or inferring structures based on protein sequences [1], is crucial in computational chemistry and material science. Traditionally, these predictions have relied on physics-inspired scoring functions [21, 11] or density functional theory (DFT)-based energy calculations [10] to define the search space, with subsequent application of search algorithms to identify optimal structures. Recently, deep generative methods, particularly diffusion models [12, 27], have proven to be highly effective in this field. These models have been successfully applied across multiple specific domains, including small molecules [34], crystals [14], antibodies [20], complexes [6], and general biomolecules [1]. The emergence of flow matching models [19], which generalize diffusion paths to more flexible probability flows, has further enhanced the generative capabilities for geometric graphs [28]. The goal of our work is to explore structure prediction from the perspective of flow matching, and align these models towards more accurate predictions.

**Aligning Generative Models.** In the domain of generative model alignment, recent work has focused on refining models to better meet human preferences. Direct Preference Optimization (DPO), introduced by [25], offers a significant advancement over traditional Reinforcement Learning from Human Feedback (RLHF, [23]) methods by directly optimizing a policy based on human preference data. This approach has proven effective in aligning large language models (LLMs) with user expectations. Extending this concept, [30] propose Diffusion-DPO, a novel method that adapts DPO for text-to-image diffusion models. By reformulating the preference optimization for diffusion model likelihoods, Diffusion-DPO achieves state-of-the-art performance in generating images that are not only visually appealing but also closely aligned with textual prompts. Recently, [36] introduces ABDPO, a DPO-based method tailored for antibody design. Unlike ABDPO, which concentrates on guiding diffusion models to generate antibody candidates with lower energy, our approach emphasizes aligning flow models for precise structure predictions.

## 3 FlowDPO

### 3.1 Flow Matching for Geometric Graphs

Flow Matching (FM, [19]) is a general paradigm for generative tasks by learning a vector field to connect the pre-defined prior distribution with the targeted data distribution. Let $q$ denote the data distribution, $x_0$ is a data point acquired from $p_0 = q$, and $x_1$ is a random sample from the prior distribution $p_1$. A time-dependent flow $\psi_t$ is then defined to shift samples from the prior distribution to the time-dependent distribution $p_t$ via the vector field $v_t$, that is

$$\psi_1(x) = x_1, \frac{d(\psi_t(x))}{dt} = v_t(\psi_t(x)). \tag{1}$$

The vector field can be further parameterized by a time-dependent model $v_\theta(x_t, t)$, leading to the continuous normalizing flows (CNFs, [5]). To avoid numerical ODE simulations to train $v_\theta$, FM simplifies the training target by aligning the model with a pre-defined vector field $u_t$ to yield $p_t$, *i.e.*,

$$\mathcal{L}_{\text{FM}} = \mathbb{E}_{t, x_t \sim p_t(x_t)}[\|v_\theta(x_t, t) - u_t(x_t)\|_2^2]. \tag{2}$$

However, as $p_t$ is still unknown, we are still unable to sample $x_t$ and apply the above objective. To address this gap, [19] leverages the more accessible conditional vector field $u_t(x_t|x_0)$ and its corresponding probability path $p_t(x_t|x_0)$, resulting in the following Conditional Flow Matching (CFM) objective, which is equivalent to $\mathcal{L}_{\text{FM}}$ in terms of gradients and accessible for sampling:

$$\mathcal{L}_{\text{CFM}} = \mathbb{E}_{t, x_t \sim p_t(x_t)}[\|v_\theta(x_t, t) - u_t(x_t|x_0)\|_2^2]. \tag{3}$$

Different vector fields lead to different probability paths. For the commonly-used Gaussian distribution defined as

$$p_t(x_t|x_0) = \mathcal{N}(x_t; \mu_t(x_0), \sigma_t^2(x_0)), \tag{4}$$

the corresponding vector field [19] is calculated as

$$u_t(x_t|x_0) = \mu'_t(x_0) + \frac{\sigma'_t(x_0)}{\sigma_t(x_0)}(x - \mu_t(x_0)), \tag{5}$$

where $\mu'_t, \sigma'_t$ are derivatives of $\mu_t, \sigma_t$ *w.r.t.* $t$. We consider three lines of Gaussian paths in this paper, namely the Variance Exploding (VE), Variance Preserving (VP) and Optimal Transport (OT) paths, which are listed in Table 1.

Table 1: Parameters of different Gaussian paths. VE, VP and OT represent Variable Exploding, Variable Preserving and Optimal Transport, respectively.

| Probability Path | Mean | Standard Deviation | Conditional Vector Field |
|---|---|---|---|
| VE path | $\mu_t(x_0) = x_0$ | $\sigma_t(x_0) = \sigma_t$ | $u_t(x_t|x_0) = \frac{\sigma'_t}{\sigma_t}(x_t - x_0)$ |
| VP path | $\mu_t(x_0) = \alpha_t x_0$ | $\sigma_t(x_0) = \sqrt{1 - \alpha_t^2}$ | $u_t(x_t|x_0) = \frac{\alpha'_t}{1-\alpha_t^2}(\alpha_t x_t - x_0)$ |
| OT path | $\mu_t(x_0) = (1 - t)x_0$ | $\sigma_t(x_0) = t$ | $u_t(x_t|x_0) = \frac{1}{t}(x_t - x_0)$ |

Based on these paths, we are capable of designing proper flow models to maintain symmetries for specific structure prediction tasks. In this paper, we mainly focus on the two typical tasks on atomic systems: antibody structure prediction and crystal structure prediction. Note that symmtries are crucial in 3D atomic systems, and we provide more discussions in Appendix B.

**Example 1: Antibody Structure Prediction.** Antibodies are Y-shaped proteins generated by the immune system to identify and bind to specific antigens, with the structure depicted in Figure 2. Researchers mainly center on the variable domains of antibodies, which comprise a heavy chain and a light chain. Each chain includes three *Complementarity-Determining Regions (CDRs)* and four framework regions in an alternating sequence. The six CDRs are volatile and crucial in defining the binding specificity and affinity, while the framework regions remain conserved. Among them, CDR-H3, which is the third CDR on the heavy chain, is the most diverse region and the primary focus of antibody design. Therefore, it is a fundamental yet challenging problem to accurately predict the structure of the CDRs upon binding.

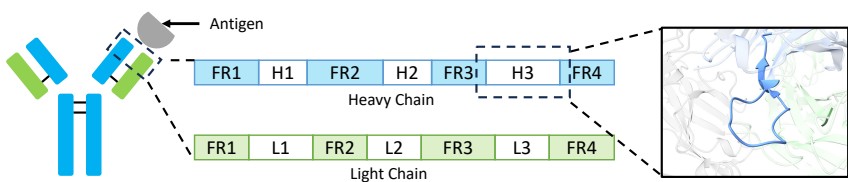

Figure 2: Graphical depiction of antibody variable domains, which consist of a heavy chain and a light chain. Each chain is equipped with 4 Framework Regions (FRs) and 3 Complementarity-Determining Regions (CDRs). The CDRs, especially CDR-H3, are volatile and thus are the key focus.

*Task Definition:* Let $\boldsymbol{A} = \{\boldsymbol{a}_1, \boldsymbol{a}_2, \cdots, \boldsymbol{a}_N\}$ denote the sequence of the targeted CDR region with the length of $N$, where $\boldsymbol{a}_i \in \{0, 1\}^{20}$ is the one-hot type of the amino acid, and $\vec{\boldsymbol{X}} = \{\vec{\boldsymbol{x}}_1, \vec{\boldsymbol{x}}_2, \cdots, \vec{\boldsymbol{x}}_N\}$ is the corresponding 3D structures with $\boldsymbol{x}_i \in \mathbb{R}^{3 \times 4}$ as the backbone coordinates including $N, C_\alpha, C,$ and $O$. Similarly, the sequence and structure of the context (*i.e.* framework regions and the antigen) are defined as $\boldsymbol{A}^C, \vec{\boldsymbol{X}}^C$. The goal is to predict the structure of the CDR region given the context:

$$\vec{\boldsymbol{X}} \sim p_0(\vec{\boldsymbol{X}}|\boldsymbol{A}, \vec{\boldsymbol{X}}^C, \boldsymbol{A}^C). \tag{6}$$

*Probability Paths and Training Objectives:* DiffAb [20] has designed the VP path for the coordinates of the CDR region as

$$\vec{\boldsymbol{u}}_{t,\text{VP}}(\vec{\boldsymbol{X}}_t|\vec{\boldsymbol{X}}_0, \boldsymbol{A}, \vec{\boldsymbol{X}}^C, \boldsymbol{A}^C) = \frac{\alpha'_t}{1 - \alpha_t^2}(\alpha_t \vec{\boldsymbol{X}}_t - \vec{\boldsymbol{X}}_0), \tag{7}$$

where $\alpha_t$ is scheduled as $\alpha_t = e^{-\frac{1}{2}\int_0^t \beta(s)ds}$. After sampling $\vec{\boldsymbol{X}}_0 = \vec{\epsilon} \sim \mathcal{N}(0, \boldsymbol{I})$, we have $\vec{\boldsymbol{X}}_t = \alpha_t \vec{\boldsymbol{X}}_0 + \sqrt{1 - \alpha_t^2}\vec{\epsilon}$. With proper reparameterization, the training objective is defined as

$$\mathcal{L}_{\text{VP}} = \mathbb{E}_{t,\vec{\epsilon}}\left[\|\vec{\epsilon}_\theta(\vec{\boldsymbol{X}}_t, \boldsymbol{A}, \vec{\boldsymbol{X}}^C, \boldsymbol{A}^C) - \vec{\epsilon}\|_2^2\right], \tag{8}$$

which only requires a model $\theta$ to predict the denoising term given the current state.

Moreover, it is also practicable to linearly connect the data point $\vec{X}_0$ and the noisy prior $\vec{\epsilon}$ via the OT path as $\vec{X}_t = (1-t)\vec{X}_0 + t\vec{\epsilon}$. The vector field is then defined as

$$\vec{u}_{t,\mathrm{OT}}(\vec{X}_t|\vec{X}_0, \boldsymbol{A}, \vec{X}^C, \boldsymbol{A}^C) = \frac{1}{t}(\vec{X}_t - \vec{X}_0) = \vec{\epsilon} - \vec{X}_0. \tag{9}$$

The training objective directly align the model with the simple vector field:

$$\mathcal{L}_{\mathrm{OT}} = \mathbb{E}_{t,\vec{\epsilon}}\big[\|\vec{v}_\theta(\vec{X}_t, \boldsymbol{A}, \vec{X}^C, \boldsymbol{A}^C) - (\vec{\epsilon} - \vec{X}_0)\|_2^2\big]. \tag{10}$$

**Example 2: Crystal Structure Prediction.** Crystal Structure Prediction (CSP), a fundamental aspect of material science, requires to predict the stable 3D structure of a compound solely from its composition. Unlike molecules or proteins, which have a finite number of atoms, the uniqueness of crystals lies in their periodic repetition in infinite 3D space. The infinite crystal structure is typically simplified by its repeating unit, which is called a *unit cell*. The key point of CSP is the representation and generation of the unit cell.

*Task Definition:* A unit cell is usually characterized by a triplet $\mathcal{M} = (\boldsymbol{A}, \boldsymbol{L}, \boldsymbol{F})$, where $\boldsymbol{A} = [\boldsymbol{a}_1, \boldsymbol{a}_2, ..., \boldsymbol{a}_N] \in \mathbb{R}^{h \times N}$ represents the one-hot encoded atom types, $\boldsymbol{L} = [\boldsymbol{l}_1, \boldsymbol{l}_2, \boldsymbol{l}_3] \in \mathbb{R}^{3 \times 3}$ denotes the lattice matrix with three basis vectors describing the crystal's periodicity, and $\boldsymbol{F} = [\boldsymbol{x}_1, \boldsymbol{x}_2, ..., \boldsymbol{x}_N] \in \mathbb{R}^{3 \times N}_{[0,1)}$ contains the fractional coordinates of the atoms, specifying their positions relative to the lattice matrix. The goal of CSP is to predict the lattice matrix and the atomic coordinates based on the given crystal composition as

$$(\boldsymbol{L}, \boldsymbol{F}) \sim p_0(\boldsymbol{L}, \boldsymbol{F}|\boldsymbol{A}). \tag{11}$$

*Probability Paths and Training Objectives:* As the lattice matrix $\boldsymbol{L}$ also lies in the Euclidean space, we can design similar VP and OT paths as Eq. (7-10). Given $\epsilon_{\boldsymbol{L}} \sim \mathcal{N}(0, \boldsymbol{I})$, with $\boldsymbol{L}_t = \alpha_t \boldsymbol{L}_0 + \sqrt{1 - \alpha_t^2}\epsilon_{\boldsymbol{L}}$, the loss function of the VP path is defined as

$$\mathcal{L}_{\boldsymbol{L},\mathrm{VP}} = \mathbb{E}_{t,\epsilon_{\boldsymbol{L}}}\big[\|\epsilon_{\boldsymbol{L},\theta}(\boldsymbol{L}_t, \boldsymbol{F}_t, \boldsymbol{A}) - \epsilon_{\boldsymbol{L}}\|_2^2\big]. \tag{12}$$

Besides, with $\boldsymbol{L}_t = (1-t)\boldsymbol{L}_0 + t\epsilon_{\boldsymbol{L}}$, the training objective of the OT path is similarly defined as

$$\mathcal{L}_{\boldsymbol{L},\mathrm{OT}} = \mathbb{E}_{t,\epsilon_{\boldsymbol{L}}}\big[\|v_{\boldsymbol{L},\theta}(\boldsymbol{L}_t, \boldsymbol{F}_t, \boldsymbol{A}) - (\epsilon_{\boldsymbol{L}} - \boldsymbol{L}_0)\|_2^2\big]. \tag{13}$$

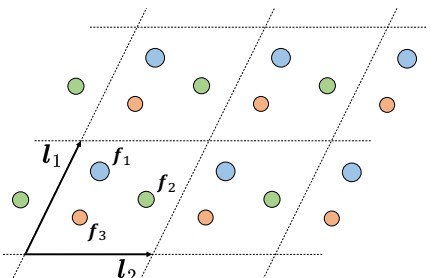

Figure 3: A crystal is the infinite periodic arrangement of atoms, and the repeating unit is named as a unit cell.

The fractional coordinates lie in the torus space of $\mathbb{R}^{3 \times N}_{[0,1)}$ to inherently reflect the periodicity of the crystal. Previous works [15, 14] project the VE path to this manifold, and the Gaussian distribution is changed into the Wrapped Normal (WN) distribution as $p_t(\boldsymbol{F}_t|\boldsymbol{F}_0) = \mathcal{N}_w(\boldsymbol{F}_t; \boldsymbol{F}_0, \sigma_t^2 \boldsymbol{I})$,

$$p_t(\boldsymbol{F}_t|\boldsymbol{F}_0) = \mathcal{N}_w(\boldsymbol{F}_t; \boldsymbol{F}_0, \sigma_t^2 \boldsymbol{I}), \tag{14}$$

where $\mathcal{N}_w(x; \cdot, \cdot) = \sum_{i=-\infty}^{\infty} \mathcal{N}_w(x + i; \cdot, \cdot)$. An accessible way to learn this path is to match the score, *i.e.* the negative logarithmic gradient, of $p_t$, and the loss function is defined as

$$\mathcal{L}_{\boldsymbol{F},\mathrm{VE}} = \mathbb{E}_{t,\boldsymbol{F}_t}\big[\lambda_t\|\epsilon_{\boldsymbol{F},\theta}(\boldsymbol{L}_t, \boldsymbol{F}_t, \boldsymbol{A}) - \nabla_{\boldsymbol{F}_t}\log p_t(\boldsymbol{F}_t|\boldsymbol{F}_0)\|_2^2\big], \tag{15}$$

where $\lambda_t = \mathbb{E}^{-1}\big[\|\nabla\log\mathcal{N}_w(0, \sigma_t^2)\|_2^2\big]$ is the pre-computed weight. If $\sigma_1$ in Eq. (14) is sufficiently large, $p_1$ would finally approach the uniform distribution, which can be selected as the prior distribution. Apart from the VE path, it is also applicable to directly connect the data point and the prior sample via the shortest path on the manifold. Specfically, given $\boldsymbol{F}_0 \sim p_0, \boldsymbol{F}_1 \sim p_1$, where $p_1$ is defined as the uniform distribution, the shortest path $s(\boldsymbol{F}_0, \boldsymbol{F}_1)$ can be determined by the logarithmic map from $\boldsymbol{F}_0$ to $\boldsymbol{F}_1$ as $s(\boldsymbol{F}_0, \boldsymbol{F}_1) = \log_{\boldsymbol{F}_0}\boldsymbol{F}_1 = w(\boldsymbol{F}_1 - \boldsymbol{F}_0 + 0.5) - 0.5$. Alternatively, $\boldsymbol{F}_1$ can also be considered as the destination of $s(\boldsymbol{F}_0, \boldsymbol{F}_1)$ via the exponential map from $\boldsymbol{F}_0$ as $\exp_{\boldsymbol{F}_0}s(\boldsymbol{F}_0, \boldsymbol{F}_1) = w(\boldsymbol{F}_0 + s(\boldsymbol{F}_0, \boldsymbol{F}_1))$. To eliminate the effect of the overall translation introduced by the prior, we further normalize $\boldsymbol{F}_1$ as $\hat{\boldsymbol{F}}_1 = \exp_{\boldsymbol{F}_0}\hat{s}(\boldsymbol{F}_0, \boldsymbol{F}_1) = \exp_{\boldsymbol{F}_0}\big(s(\boldsymbol{F}_0, \boldsymbol{F}_1) - \bar{s}(\boldsymbol{F}_0, \boldsymbol{F}_1)\big),$

where $\bar{s}$ averages the paths of all atoms. With the path of $\boldsymbol{F}_t$ defined as $\boldsymbol{F}_t = \exp_{\boldsymbol{F}_0}\left(t\hat{s}(\boldsymbol{F}_0, \boldsymbol{F}_1)\right)$, the training objective for the OT path is

$$\mathcal{L}_{\boldsymbol{F},\text{OT}} = \mathbb{E}_{t,\boldsymbol{F}_1}\left[\|\boldsymbol{v}_{\boldsymbol{F},\theta}(\boldsymbol{L}_t, \boldsymbol{F}_t, \boldsymbol{A}) - \hat{s}(\boldsymbol{F}_0, \boldsymbol{F}_1)\|_2^2\right]. \tag{16}$$

**Generalized Notations.** Overall, the structure prediction tasks aims at generating the targeted structure $x$ given some condition $c$, *i.e.* to learn $p_0(x|c)$. And the flow matching objective minimizes the Mean Square Error (MSE) of the predicted and pre-defined vector fields with proper reparameterization or simplification, which can be generalized as

$$\mathcal{L} = \mathbb{E}_{t,x_0\sim p_0,x_1\sim p_1}[\text{MSE}_t(x_0, x_1; \theta)]. \tag{17}$$

Hereinafter, we use these generalized notations for simplicity.

## 3.2 Preference Dataset Construction

Building on the flow paths introduced in § 3.1, we now delve into the details of constructing a preference dataset, which is pivotal for the application of DPO, as detailed in § 3.3.

---

**Algorithm 1** Candidate Generation

1: **Input:** $N$, $M$, $\mathcal{D}_{ref} = \{(x_i, c_i)\}_{i=1}^N$, $\theta_{\text{ref}}$, $d(\cdot,\cdot)$, $\delta$
2: **Output:** $\mathcal{D}_{gen}, \mathcal{D}_{pos}, check$
3: **Initialize:** $\mathcal{D}_{gen}, \mathcal{D}_{pos}, check \leftarrow [], [], []$
4: **for** $i = 1$ to $N$ **do**
5:     $\mathcal{D}_{gen}[i], \mathcal{D}_{pos}[i] \leftarrow [], []$
6:     $match \leftarrow$ False, $j_{pos} \leftarrow 1$
7:     **for** $j = 1$ to $M$ **do**
8:        Generate $\hat{x}_{ij} \sim p(x|c_i; \theta_{\text{ref}})$
9:        $\mathcal{D}_{gen}[i, j] \leftarrow \hat{x}_{ij}$
10:       **if** $d(x_i, \hat{x}_{ij}) \leq \delta$ **then**
11:         $\mathcal{D}_{pos}[i, j_{pos}] \leftarrow \hat{x}_{ij}$
12:         $match \leftarrow$ True, $j_{pos} \leftarrow j_{pos} + 1$
13:       **end if**
14:     **end for**
15:     $check[i] \leftarrow match$
16: **end for**

---

**Algorithm 2** Preference Pairs Construction

1: **Input:** $\mathcal{D}_{ref}$, $\mathcal{D}_{gen}$, $\mathcal{D}_{pos}$, $check$, $N$, $K$, $d(\cdot,\cdot)$, $r$
2: **Output:** $\mathcal{D}_{pair}$
3: **Initialize:** $\mathcal{D}_{pair} \leftarrow []$
4: **for** $i = 1$ to $N$ **do**
5:     $\mathcal{D}_{pair}[i] \leftarrow []$
6:     **for** $k = 1$ to $K$ **do**
7:        **if** $k \leq rK$ **or** $check[i] =$ False **then**
8:         $x_{ik}^w \leftarrow x_i$
9:         $x_{ik}^l \sim \mathcal{D}_{gen,i}$
10:       **else**
11:         $x_{ik}^w \sim \mathcal{D}_{pos,i}$, $x_{ik}^l \sim \mathcal{D}_{gen,i}$
12:         Swap if $d(x_i, x_{ik}^w) > d(x_i, x_{ik}^l)$
13:       **end if**
14:       $\mathcal{D}_{pair}[i, j] \leftarrow (x_{ik}^w, x_{ik}^l, c_i)$
15:     **end for**
16: **end for**

---

**Candidate Generation** As shown in Algorithm 1, the construction of the preference dataset begins with the generation of multiple candidate structures for each sample in our reference dataset, $\mathcal{D}_{ref}$. Leveraging the pre-trained flow-based generative model $\theta_{ref}$, we generate $M$ candidate structures $\{\hat{x}_{ij}\}_{j=1}^M$ for each sample $(x_i, c_i)$ via $p(x|c_i; \theta_{ref})$, ensuring that each generated structure is contextually relevant and adheres to the geometric constraints discussed previously.

As each candidate is generated, we compute the distance between $\hat{x}_{ij}$ and the original structure $x_i$ using a predefined metric $d(\cdot, \cdot)$. If this distance is less than or equal to a threshold $\delta$, the candidate is considered a close match and is added to $\mathcal{D}_{pos}$, a subset of promising candidates. This step is crucial for efficiently filtering the generated data to retain only the most relevant candidates for DPO.

**Preference Pairs Construction** Subsequently, we construct $K$ preference pairs $(x_{ik}^w, x_{ik}^l)$ for each sample $i$ by Algorithm 2, where $x_{ik}^w$ is preferred over $x_{ik}^l$. This preference is determined based on their proximity to the original structure $x_i$. Apart from sampling pairs from generated structures, we also use a ratio $r$ to select the ground truth as the preferred sample. Moreover, if all generated structures for a sample are far from the original, the original structure $x_i$ is always preferred. The other pairs are formed by selecting $x_{ik}^w$ from the promising subset $\mathcal{D}_{pos}$ and $x_{ik}^l$ from the broader set $\mathcal{D}_{gen}$. This process ensures that the pairs reflect a clear preference based on the closeness to the original structure, facilitating effective training through DPO, which is explored in the next section.

## 3.3 Direct Preference Optimization

To align large language models with human preference, DPO [25] is proposed to replace the RLHF [23] training objective with directly maximizing the likelihood of the preference. [30] extends DPO to text-to-image generation task, adapting the DPO target to diffusion models. Given the preference pair $(x^w, x^l)$, DPO [25] designs the training objective as

$$\mathcal{L}_{\text{DPO}} = -\mathbb{E}_{x^w, x^l} \Big[ \log \sigma \big( \beta \log \frac{p_{\text{opt}}(x^w)}{p_{\text{ref}}(x^w)} - \beta \log \frac{p_{\text{opt}}(x^l)}{p_{\text{ref}}(x^l)} \big) \Big], \tag{18}$$

where $p_{\text{opt}}, p_{\text{ref}}$ are probabilities yielded by the fine-tuned model $\theta_{\text{opt}}$ and the pre-trained flow model, and $\beta$ is a hyperparameter to control the KL divergence of these two distributions.

It is nontrivial to efficiently acquire $p(x)$ via iterative generative models. Inspired by [30], we uniformly discretize the time interval into $T$ steps, where step $i$ is located at $t = i/T$. By formulating the probability from the path $x_{0:T}$, Eq. (18) can be rewritten as

$$\mathcal{L}_{\text{DPO}} = -\mathbb{E}_{x^w, x^l} \log \sigma \Big( \beta \mathbb{E}_{x^w_{1:T}, x^l_{1:T}} \big[ \log \frac{p_{\text{opt}}(x^w_{0:T})}{p_{\text{ref}}(x^w_{0:T})} - \log \frac{p_{\text{opt}}(x^l_{0:T})}{p_{\text{ref}}(x^l_{0:T})} \big] \Big). \tag{19}$$

To avoid costly sampling through the entire path, Jensen's inequality [30] is applied to bound Eq. (19) as

$$\mathcal{L}_{\text{DPO}} \le -\mathbb{E}_{x^w, x^l, i} \log \sigma \Big( B \big[ \log \frac{p_{\text{opt}}(x^w_{i-1}|x^w_i)}{p_{\text{ref}}(x^w_{i-1}|x^w_i)} - \log \frac{p_{\text{opt}}(x^l_{i-1}|x^l_i)}{p_{\text{ref}}(x^l_{i-1}|x^l_i)} \big] \Big), \tag{20}$$

where $B = \beta T$ servers as a hyperparameter. As directly sampling $x_{i-1}, x_i$ from an arbitrary intermediate step $i$ is still unfeasible, we can estimate them via the accessible Gaussian paths $p$ in Table 1 as

$$\mathcal{L}_{\text{DPO}} = -\mathbb{E}_{x^w, x^l, i} \log \sigma \Big( B \mathbb{E}_{p(x^w_{i-1}|x^w_{i,0}), p(x^w_{i-1}|x^l_{i,0})} \big[ \log \frac{p_{\text{opt}}(x^w_{i-1}|x^w_i)}{p_{\text{ref}}(x^w_{i-1}|x^w_i)} - \log \frac{p_{\text{opt}}(x^l_{i-1}|x^l_i)}{p_{\text{ref}}(x^l_{i-1}|x^l_i)} \big] \Big) \tag{21}$$

$$= -\mathbb{E}_{x^w, x^l, i} \log \sigma \Big( B \big[ \mathcal{J}(x^w_i; p, p_{\text{ref}}) - \mathcal{J}(x^w_i; p, p_{\text{opt}}) - \mathcal{J}(x^l_i; p, p_{\text{ref}}) + \mathcal{J}(x^l_i; p, p_{\text{opt}}) \big] \Big), \tag{22}$$

where $\mathcal{J}(x^w_i; p, p_\theta)$ denotes $D_{\text{KL}}\big( p(x^w_{i-1}|x^w_{i,0}) \| p_\theta(x^w_{i-1}|x^w_i) \big)$ and the same for $\mathcal{J}(x^l_i; p, p_\theta)$. As $p$ and $p_\theta$ are Gaussian distributions with the same noise scheduler, the KL divergence can be formulated as

$$\mathcal{J}(x_i; p, p_\theta) = \frac{1}{2\sigma^2_{i-1|i}} \Big\| \mu(x_{i-1}|x_{i,0}) - \mu_\theta(x_{i-1}|x_i) \Big\|^2_2. \tag{23}$$

According to DDIM [27], if a time-dependent Gaussian path follows the form $x_i \sim \mathcal{N}(x_i; k_i x_0, \sigma_i \boldsymbol{I})$, we can further design $p(x_{i-1}|x_{i,0}) = \mathcal{N}\big(x; \mu(x_{i-1}|x_{i,0}), \sigma^2_{i-1|i}\big)$. Given $\sigma^2_{i-1|i}$, the mean can be formulated as

$$\mu(x_{i-1}|x_{i,0}) = \frac{1}{\sigma_i} \sqrt{\sigma^2_{i-1} - \sigma^2_{i-1|i}} x_i + \Big( k_{i-1} - \frac{k_i}{\sigma_i} \sqrt{\sigma^2_{i-1} - \sigma^2_{i-1|i}} \Big) x_0. \tag{24}$$

Fortunately, all paths defined in Table 1 follows this form. And $\mu_\theta(x_{i-1}|x_i)$ can be parameterized similarly as Eq. (24), with estimating $x_0$ via predicted vector field or denoising terms. Hence, we can approximate $\mathcal{J}(x_i; p, p_\theta)$ by $\text{MSE}_i(x_0, x_1; \theta)$. With sufficiently large $T$, Eq. (22) can be changed into an applicable form as follows, which is our final training objective.

$$\mathcal{L}_{\text{DPO}} = -\mathbb{E}_{x^w_{0,1}, x^l_{0,1}, t} \log \sigma \Big( B \big[ \text{MSE}_t(x^w_0, x^w_1; \theta_{\text{ref}}) - \text{MSE}_t(x^w_0, x^w_1; \theta_{\text{opt}})$$
$$- \text{MSE}_t(x^l_0, x^l_1; \theta_{\text{ref}}) + \text{MSE}_t(x^l_0, x^l_1; \theta_{\text{opt}}) \big] \Big), \tag{25}$$

Table 2: $C_\alpha$ and bb indicates RMSD calculated on $C_\alpha$ atoms and backbone atoms, repectively. $C_\alpha$-w and bb-w averages the RMSDs of the worst generated conformations of each complex.

| Model | L1 | | | | L2 | | | | L3 | | | |
|---|---|---|---|---|---|---|---|---|---|---|---|---|
| | $C_\alpha$-w | $C_\alpha$ | bb-w | bb | $C_\alpha$-w | $C_\alpha$ | bb-w | bb | $C_\alpha$-w | $C_\alpha$ | bb-w | bb |
| VP Path [20] | 2.71 | 2.00 | 2.56 | 2.06 | 1.11 | 0.95 | 1.08 | 0.96 | 1.32 | 0.99 | 1.39 | 1.08 |
| OT Path | 2.25 | 1.77 | 2.24 | 1.83 | 1.13 | 0.96 | 1.10 | 0.96 | 1.49 | 1.05 | 1.45 | 1.13 |
| VP Path + DPO | 2.47 | 1.91 | 2.31 | 1.95 | **1.09** | 0.94 | 1.07 | 0.94 | **1.22** | **0.94** | **1.30** | **1.01** |
| OT Path + DPO | **2.22** | **1.74** | **2.19** | **1.78** | 1.09 | **0.93** | **1.05** | **0.93** | 1.28 | 0.95 | 1.34 | 1.05 |

| Model | H1 | | | | H2 | | | | H3 | | | |
|---|---|---|---|---|---|---|---|---|---|---|---|---|
| | $C_\alpha$-w | $C_\alpha$ | bb-w | bb | $C_\alpha$-w | $C_\alpha$ | bb-w | bb | $C_\alpha$-w | $C_\alpha$ | bb-w | bb |
| VP Path [20] | 1.18 | 0.83 | 1.14 | 0.89 | 1.41 | 0.92 | 1.45 | 1.00 | 5.01 | 3.77 | 4.95 | 3.78 |
| OT Path | 1.31 | 0.89 | 1.26 | 0.94 | 1.69 | 1.06 | 1.60 | 1.13 | 4.81 | 3.66 | 4.83 | 3.70 |
| VP Path + DPO | **1.13** | **0.80** | **1.13** | **0.86** | **1.35** | **0.87** | **1.37** | **0.95** | 4.42 | 3.44 | 4.38 | 3.45 |
| OT Path + DPO | 1.23 | 0.83 | 1.19 | 0.89 | 1.46 | 0.95 | 1.41 | 1.02 | **4.28** | **3.32** | **4.23** | **3.32** |

## 4 Experiments

We validate our method on two distinct domains: antibody structure prediction (§ 4.1) and crystal structure prediction (§ 4.2).

### 4.1 Antibody Structure Prediction

**Dataset** Following previous literature [20], we extract antibody structures from the SAbDab database [8] for training and utilize the manually curated test set from DiffAb [20], which contains 19 antibody-antigen complexes. We first derive all structures deposited before April 11th, 2024, and remove those with resolution above 4.0Å or non-protein targets, resulting in 12,428 antibodies. Subsequently, we use mmseqs2 [29] to cluster the antibodies based on 50% sequence identity for each CDR, and exclude those in the same clusters as the test set antibodies. The dataset is then split into training and validation sets at a 9:1 ratio based on the clusters.

**Metrics** We employ the following metrics for evaluation. $\mathbf{RMSD}_{C_\alpha}$ measures the Root Mean Square Deviation of the generated alpha carbon coordinates with respect to the reference. $\mathbf{RMSD}_{bb}$ is the RMSD calculated on the four backbone atoms including $C, C_\alpha, N, O$. To better profile the generated distribution, for each antibody, we generate 20 structures and use two strategies to aggregate the results across different antibodies. Strategy **worst** select the worst generated structure per antibody according to RMSD and then averagse across different antibodies, while strategy **mean** averages the RMSD of 20 candidates first, and then across antibodies. Strategy worst measures the furthest deviation of the generated distribution compared to the reference, while strategy mean is commonly adopted in previous works [20, 16]. Results aggregated with **worst** are denoted as $\mathbf{C_\alpha}$**-w** and **bb-w**, while those with **mean** are denoted as $\mathbf{C_\alpha}$ and **bb**.

**Results** We evaluate VP path (DiffAb) [20] and OT path [19] with the proposed FlowDPO on CDR structure prediction. Results in Table 2 illustrate that either using VP path or OT path, further training with DPO consistently enhances performance across different CDRs. Notably, on the most challenging part (*i.e.* CDR-H3), the DPO phase yields the most significant improvement. Metrics aggregated with strategy **worst** demonstrate noticeable gains, indicating effective supperssion of low-quality samples by the DPO phase, which we attribute to the objective of DPO in distinguishing the prefered samples. Such characteristics are favorable in practical applications where it requires blind selection of generated structures without prior knowledge of which structures might be more correct. We also depict the distributions of RMSD and examples of generated CDR-H3 structures in Figure 4. It shows that the blue curves, yielded by the original flow models, often exhibit a bimodal distribution. While the first peak at a lower RMSD indicating higher quality generations, the second peak at a higher RMSD suggests the models experience *hallucination*, confidently generating conformations that significantly deviate from the ground truth. DPO effectively suppresses this erroneous second peak, leading to an overall improvement in the quality of generated samples. On closer inspection, this correction also addresses physical invalidities, such as the twisted backbone seen in Figure 4.

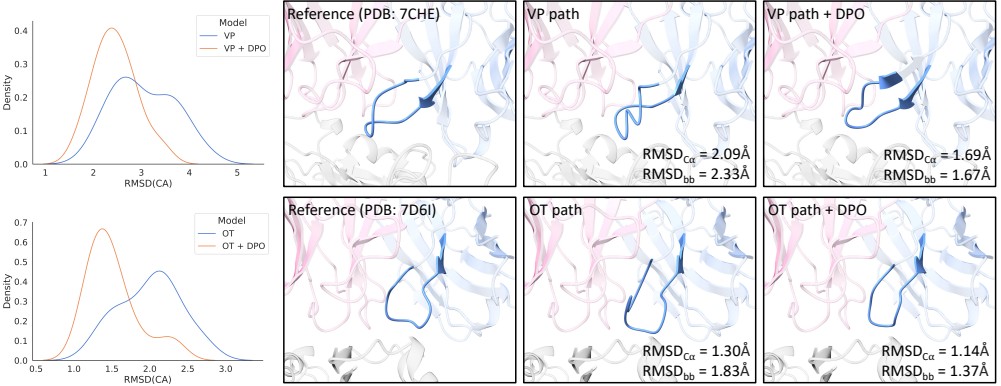

Figure 4: Examples of generated CDR-H3 structures and the distribution of $RMSD_{C_\alpha}$ for different antigen-antibody complexes and different probability paths. The visualized samples are the ones with the lowest RMSD of all the generated counterparts for the corresponding complexes. In addition to driving the distribution towards lower RMSD, it is also observed that the DPO phase tends to rectify the physical invalidity (*e.g.* twisted backbone in the above examples) in the generated samples.

## 4.2 Crystal Structure Prediction

**Dataset** We conduct the crystal structure prediction task on three datasets in line with previous works [33, 14]. **Perov-5** [4] includes 18,928 perovskite crystals, each characterized by similar structures but varying compositions, and exactly 5 atoms per unit cell. **MP-20** [13] comprises 45,231 materials from the Materials Project, featuring a wide range of compositions and structures, with each material containing no more than 20 atoms per unit cell. These materials predominantly represent crystals that have been synthesized experimentally. **MPTS-52** is an advanced version of MP-20, containing 40,476 structures with unit cells that include up to 52 atoms, presenting a more complex challenge. For Perov-5 and MP-20, we maintain the conventional 60-20-20 split for training, validation, and testing. For the MPTS-52 dataset, we use a chronological split, assigning 27,380 crystals for training, 5,000 for validation, and 8,096 for testing.

**Metrics** For inference, we generate one structure given each composition. The predicted sample is then matched with the ground truth via the StructureMatcher class in pymatgen [22] with thresholds stol=0.5, angle_tol=10, ltol=0.3 as applied in previous works [33, 14]. We use **Match Rate (MR)** as the proportion of matched structures among the testing set, and the **RMSD** is averaged over the matched pairs, and normalized by $\sqrt[3]{V/N}$ where $V$ is the volume of the unit cell.

**Results** We compare the results with two generative baselines **P-cG-SchNet** [9] and **CDVAE** [33]. The results are shown in Table 3, where we explore three combinations of paths for jointly generating the lattice and the fractional coordinates: VP+VE, OT+OT, and OT+VE. Notably, the VP+VE path is previously developed by DiffCSP [14]. We find that the OT path is more effective for lattice generation, while the VE path provides more accurate predictions of atomic coordinates within the cell. Overall, the OT+VE combination generally delivers the best performance. Furthermore, DPO consistently enhances the performance of the model trained on each combination, demonstrating its capability to refine the predictions to a more precise alignment with experimental structures. We additionally visualize the RMSD distribution of predicted structures from different Gaussian paths. Results in Figure 5 reveal a similar pattern to Figure 4, demonstrating that DPO reduces the probability of low-quality generations.

## 5 Conclusion

In this work, we propose FlowDPO, a novel framework for 3D structure prediction that integrates flow-based generative models with Direct Preference Optimization. We achieve 3D structure prediction via flow matching models with various probability paths, and generalize the DPO training objective to arbitrary Gaussian paths. To refine the model via DPO, we generate multiple candidate structures and construct the preference dataset by aligning with ground truth. The results demonstrate substantial

Table 3: Results on crystal structure prediction task. MR stands for Match Rate.

| | Perov-5 | | MP-20 | | MPTS-52 | |
|---|---|---|---|---|---|---|
| | MR (%) | RMSE | MR (%) | RMSE | MR (%) | RMSE |
| P-cG-SchNet [9] | 48.22 | 0.4179 | 15.39 | 0.3762 | 3.67 | 0.4115 |
| CDVAE [33] | 45.31 | 0.1138 | 33.90 | 0.1045 | 5.34 | 0.2106 |
| VP + VE Path [14] | 52.02 | **0.0760** | 51.49 | 0.0631 | 12.19 | 0.1786 |
| OT + OT Path | 53.95 | 0.1508 | 57.40 | 0.1185 | 17.40 | 0.2405 |
| OT + VE Path | 52.29 | 0.0782 | 58.94 | 0.0621 | 18.91 | 0.1435 |
| VP + VE + DPO | 53.47 | 0.0762 | 59.98 | 0.0622 | 14.75 | 0.1780 |
| OT + OT + DPO | **55.56** | 0.1376 | 59.62 | 0.0898 | **22.36** | 0.1678 |
| OT + VE + DPO | 53.94 | 0.0765 | **62.47** | **0.0606** | 20.27 | **0.1419** |

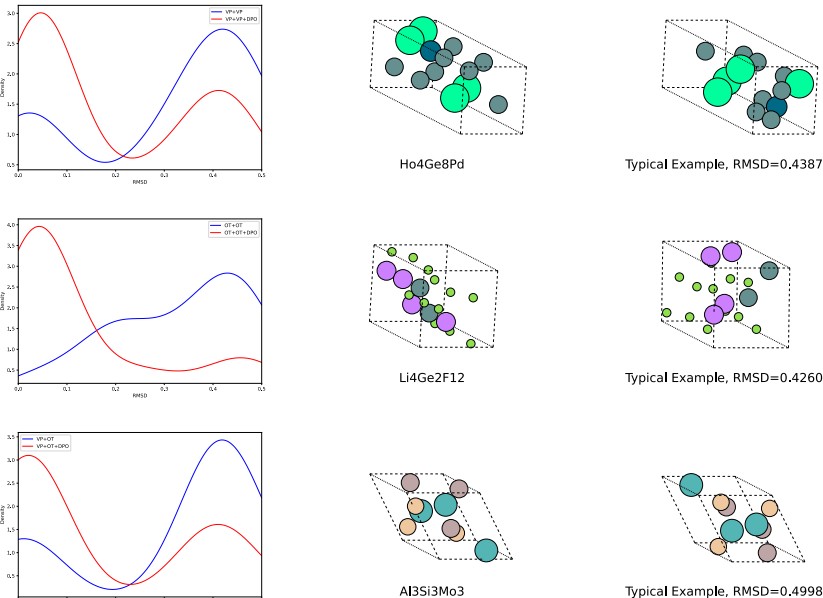

Figure 5: Visualizations on crystal structure prediction results. The left column depicts the RMSD distribution of the models before (blue) and after (red) DPO. The middle column shows the ground truth structures, and the right column shows typical high RMSD generations to be suppressed.

improvements in prediction accuracy for both antibody and crystal structures, highlighting the effectiveness and versatility of FlowDPO in the field of 3D structure prediction.

## Acknowledgments

This work is jointly supported by the National Science and Technology Major Project under Grant 2020AAA0107300, the National Natural Science Foundation of China (No. 61925601, No. 62376276, No. 62236011), and Beijing Nova Program (20230484278).

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

# A  From RLHF to DPO

Given the preference pair $(x^w, x^l)$ with condition $c$, the Bradley-Terry (BT) model considers a latent reward model $r(x|c)$ behind them and formulates the preference as

$$p(x^w \succ x^l|c) = \frac{\exp(r(x^w|c))}{\exp(r(x^w|c)) + \exp(r(x^l|c))}. \tag{26}$$

RLHF [23] optimizes the generative model by explicitly training a reward model $r_\phi$, and maximizing the reward with a KL regularization term to control the model by the original reference $p_{\text{ref}}$ as

$$\max_{p_{\text{opt}}} \mathbb{E}_{x \sim p_{\text{opt}}(x)}[r_\phi(x)] - \beta D_{\text{KL}}[p_{\text{opt}}(x) \| p_{\text{ref}}(x)]. \tag{27}$$

We omit the condition $c$ for simplicity. As Eq. (27) exists a close-form solution $p_\theta^*(x) = p_{\text{ref}}(x)e^{r^*(x)/\beta}/Z$, where $Z$ is the normalization term, we can rewrite the optimal reward model as

$$r^*(x) = \beta \log \frac{p_{\text{opt}}(x)}{p_{\text{ref}}(x)} + \beta Z. \tag{28}$$

After introducing Eq. (28) into Eq. (26) and directly maximizing the log likelihood, DPO [25] simplifies the training objective as

$$\mathcal{L}_{\text{DPO}} = -\mathbb{E}_{x^w, x^l}\left[ \log \sigma\left(\beta \log \frac{p_{\text{opt}}(x^w)}{p_{\text{ref}}(x^w)} - \beta \log \frac{p_{\text{opt}}(x^l)}{p_{\text{ref}}(x^l)}\right)\right]. \tag{29}$$

# B  Required Symmetries of Atomic Systems

The design of flow paths is constrained by the symmetry requirements of specific atomic systems, which are detailed as follows.

**Antibody Structure Prediction**  The designed vector field should maintain equivariance to any rotation $Q \in SO(3)$ and be invariant to any translation $\vec{t} \in \mathbb{R}^3$:

$$\vec{u}_t(Q\vec{X}_t + \vec{t}|Q\vec{X}_0 + \vec{t}, A, Q\vec{X}^C + \vec{t}, A^C) = Q\vec{u}_t(\vec{X}_t|\vec{X}_0, A, \vec{X}^C, A^C). \tag{30}$$

**Crystal Structure Prediction**  Previous works [14, 35] consider the task defined in Eq. (11) as a joint generation task on $L$ and $F$. For the generative process, the vector field of the lattice should be equivariant to an arbitrary rotation $Q \in SO(3)$, and that of the coordinates is required to ensure the periodic translation invariance for any translation vector $t$. Specfically, we have

$$u_{L,t}(QL_t|QL_0, F_0, A) = Qu(L_t|L_0, F_0, A), \tag{31}$$
$$u_{F,t}(w(F_t + t)|L_0, w(F_0 + t), A) = u(L_t|L_0, F_0, A), \tag{32}$$

where the operation $w(F) = F - \lfloor F \rfloor \in [0, 1)^{3 \times N}$ returns the coordinates back to the unit cell.

After maintaining the required symmetries, the proposed flow model is capable of generating equivalent structures under different E(3) transformations. An example of the OT-OT path for the crystal is shown in Figure 6.

# C  Implementation Details

## C.1  Antibody Structure Prediction

We use the framework of DiffAb [20] to train the flow models. The original denoising network in DiffAb requires orientation matrices as input, yet the OT path of the $SO(3)$ matrices is not naive to derive, which is out of our main scope. Therefore, we replace the denoising network with the multi-channel EGNN proposed in MEAN [16] to avoid this problem. All experiments can be run on one GeForce RTX 3090 GPU. Detailed hyperparameters for our FlowDPO are presented in Table 4.

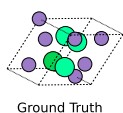 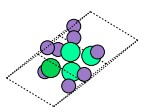 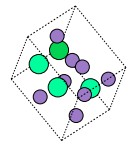 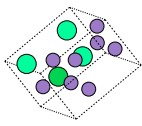 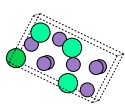

Ground Truth                                        Generated Samples

Figure 6: Visualizations of multiple generated crystals via OT-OT path on MP-20. As the designed path maintain the symmetries, the model is able to generate structures equivalent to the ground truth after proper rotations and (periodic) translations.

Table 4: Hyperparameters for the antibody structure prediction task.

|  | CDR | Flow | | | | Preference Dataset | | | | DPO | | |
|---|---|---|---|---|---|---|---|---|---|---|---|---|
|  |  | d | L | Lr | Epoch | M | $K^2$ | $\delta$ | r | Lr | Epoch | B |
| VP | L1 | 128 | 6 | 1e-4 | 500 | 5 | 1 | 1.0 | 0.1 | 5e-6 | 50 | 200 |
|  | L2 | 128 | 6 | 1e-4 | 400 | 5 | 1 | 1.0 | 0.1 | 1e-6 | 20 | 200 |
|  | L3 | 128 | 6 | 1e-4 | 500 | 5 | 1 | 1.0 | 0.1 | 5e-6 | 5 | 200 |
|  | H1 | 128 | 6 | 1e-4 | 500 | 5 | 1 | 1.0 | 0.1 | 3e-6 | 10 | 200 |
|  | H2 | 128 | 6 | 1e-4 | 500 | 5 | 1 | 1.0 | 0.1 | 5e-6 | 10 | 200 |
|  | H3 | 128 | 6 | 1e-4 | 500 | 5 | 1 | 1.0 | 0.1 | 5e-6 | 5 | 200 |
| OT | L1 | 128 | 6 | 1e-4 | 500 | 5 | 1 | 1.0 | 0.1 | 5e-6 | 50 | 200 |
|  | L2 | 128 | 6 | 1e-4 | 500 | 5 | 1 | 1.0 | 0.05 | 5e-6 | 20 | 200 |
|  | L3 | 128 | 6 | 1e-4 | 900 | 5 | 1 | 1.0 | 0.1 | 5e-6 | 5 | 200 |
|  | H1 | 128 | 6 | 1e-4 | 900 | 5 | 1 | 1.0 | 0.0 | 5e-6 | 25 | 200 |
|  | H2 | 128 | 6 | 1e-4 | 700 | 5 | 1 | 1.0 | 0.1 | 5e-6 | 50 | 200 |
|  | H3 | 128 | 6 | 1e-4 | 500 | 5 | 1 | 1.0 | 0.1 | 5e-6 | 5 | 200 |

### C.2 Crystal Structure Prediction

We use the denoising network designed in DiffCSP [14] as the backbone model to train the flow models. To predict the vector field via the denoising output, for the OT path designed on lattice, we apply the reparameterization as

$$
v_{L,\theta}(L_t, F_t, A) = \begin{cases} 0, t = 1, \\ \dfrac{\epsilon_{L,\theta}(L_t, F_t, A) - L_t}{1 - t}, 0 \le t < 1. \end{cases}
\tag{33}
$$

And for the OT path on the fractional coordinates, we directly use $v_{F,\theta}(L_t, F_t, A) = \epsilon_{F,\theta}(L_t, F_t, A)$. We select RMSD defined by StructureMatcher class in pymatgen [22] with thresholds stol=0.5, angle_tol=10, ltol=0.3 as the distance metric to construct the preference dataset. Specially, the RMSD of the unmatched structure is set as $+\infty$, and such candidates will never be selected as the preferred sample.

The detailed hyperparameters for the FlowDPO pipeline on each crystal dataset are provided in Table 5. Each experiment is run on one GeForce RTX 3090 GPU.

## D   Comparison with Regressive Methods

To further investigate the advantages of the generative paradigm, we employ the same backbone model (MEAN) for a direct regression task as an additional baseline. The results are presented in Table 6. Our findings indicate that generative models outperform the regression model in 4 of the 6 CDRs, particularly in the highly variable and functionally critical regions, CDR-H3 and CDR-L3.

---

[2]Each time the pair is randomly sampled from the 5 candidates plus the ground truth.

Table 5: Hyperparameters for the crystal structure prediction task.

| | | Flow | | | | Preference Dataset | | | | | DPO | |
| | | d | L | Lr | Epoch | M | K | $\delta$ | r | Lr | Epoch | B |
|---|---|---|---|---|---|---|---|---|---|---|---|---|
| VP+VE | Perov-5 | 256 | 4 | 1e-3 | 3000 | 5 | 12 | 0.3 | 1/6 | 1e-3 | 100 | 2000 |
| | MP-20 | 512 | 6 | 1e-3 | 1000 | 5 | 12 | 0.3 | 1/6 | 1e-3 | 300 | 2000 |
| | MPTS-52 | 512 | 6 | 1e-3 | 1000 | 5 | 12 | 0.3 | 1/6 | 1e-3 | 200 | 2000 |
| OT+OT | Perov-5 | 256 | 4 | 1e-3 | 3000 | 5 | 12 | 0.3 | 1/6 | 1e-4 | 70 | 2000 |
| | MP-20 | 512 | 6 | 1e-3 | 3000 | 5 | 12 | 0.3 | 1/6 | 1e-4 | 30 | 2000 |
| | MPTS-52 | 512 | 6 | 1e-3 | 2000 | 5 | 12 | 0.3 | 1/6 | 1e-4 | 100 | 2000 |
| OT+VE | Perov-5 | 256 | 4 | 1e-3 | 3000 | 5 | 12 | 0.3 | 1/6 | 5e-5 | 5 | 2000 |
| | MP-20 | 512 | 6 | 1e-3 | 3000 | 5 | 12 | 0.3 | 1/6 | 1e-3 | 300 | 2000 |
| | MPTS-52 | 512 | 6 | 1e-3 | 2000 | 5 | 12 | 0.3 | 1/6 | 1e-3 | 200 | 2000 |

Additionally, we report both the mean and minimum RMSD values across 20 generations for each generative model. The significantly lower minimum RMSD values demonstrate that generative models not only yield predictions closer to the observed reference structures but also possess the capability to generate multiple feasible conformations around the stable state.

Table 6: Results compare to regressive baselines on antibody structure prediction tasks.

| CDR | Regression | | FlowDPO (VP, mean) | | FlowDPO (OT, mean) | | FlowDPO (VP, min) | | FlowDPO (OT, min) | |
|---|---|---|---|---|---|---|---|---|---|---|
| | $RMSD_{CA}$ | $RMSD_{bb}$ | $RMSD_{CA}$ | $RMSD_{bb}$ | $RMSD_{CA}$ | $RMSD_{bb}$ | $RMSD_{CA}$ | $RMSD_{bb}$ | $RMSD_{CA}$ | $RMSD_{bb}$ |
| L1 | 1.03 | 1.05 | 1.91 | 1.95 | 1.74 | 1.78 | 1.44 | 1.60 | 1.22 | 1.33 |
| L2 | 0.96 | 0.92 | 0.94 | 0.94 | 0.93 | 0.93 | 0.80 | 0.85 | 0.83 | 0.87 |
| L3 | 1.17 | 1.18 | 0.94 | 1.01 | 0.95 | 1.05 | 0.69 | 0.82 | 0.69 | 0.84 |
| H1 | 1.68 | 1.67 | 0.80 | 0.86 | 0.83 | 0.89 | 0.53 | 0.67 | 0.52 | 0.67 |
| H2 | 0.72 | 0.78 | 0.87 | 0.95 | 0.95 | 1.02 | 0.49 | 0.65 | 0.57 | 0.71 |
| H3 | 3.46 | 3.48 | 3.44 | 3.45 | 3.32 | 3.32 | 2.60 | 2.64 | 2.55 | 2.61 |

## E  Limitations

As detailed in § 3.3, the derivation of the rationality of DPO for flow models primarily focuses on Gaussian paths. However, flow models have the potential to learn mappings from an arbitrary prior to the data distribution if the probability path is appropriately defined. Therefore, a more general derivation that does not rely on Gaussian assumptions could be explored in future research. Additionally, our current evaluation is based predominantly on computational metrics. Conducting wet-lab experiments would provide a more robust validation of the model's effectiveness in practical applications.

## F  Broader Impacts

The introduction of FlowDPO marks a pivotal advancement in scientific domains such as drug development, materials research, and molecular informatics. Recent developments, such as AlphaFold3, have demonstrated remarkable accuracy in predicting structures across various domains [1]. However, issues such as hallucinations, like erroneous structural order in inherently disordered regions, remain a challenge. It is intriguing to explore whether alignment strategies based on DPO can mitigate these hallucinations and enhance overall prediction accuracy.

## G  Codes

Our codes are available at `https://github.com/jiaor17/FlowDPO`.

