# OpenReview forum: "3D Structure Prediction of Atomic Systems with Flow-based Direct Preference Optimization"
_NeurIPS.cc/2024/Conference — NeurIPS 2024 poster_

### Official Review · Reviewer_jaPH · 2024-07-09

**Soundness:** 3
**Presentation:** 3
**Contribution:** 3
**Rating:** 6
**Confidence:** 2

**Summary:**

The paper proposes to apply the Direct Preference Optimization (DPO) procedure for the 3D Structure Prediction of atomic structure predictions. The approach is applied to crystal and antibody structures, considering different kinds of Gaussian Path for flow matching models, showing a general improvement in the state of the art.

**Strengths:**

- The method seems to consistently outperform previous methods on the two different tasks
- The implementation of the code approach can foster the application of DPO protocol to other problems in the chemical and molecular domains.
- Experimentations with different kinds of Gaussian paths seem to be an interesting analysis that could help future works to build on top of them.

I may have missed some important aspects since I am not an expert in this domain.

**Weaknesses:**

- As a non-expert, I have a hard time understanding some of the merits. In particular, the approach seems mainly to be a straightforward application of DPO to the specific case of atomic systems, and I cannot find a technical contribution specifically designed for the specific case. Given that I do not know how relevant the problems considered are and what their impact could be, I am unsure about the significance of the work. In case I am missing something, I am looking forward to authors' clarification.
- I feel the general presentation could be improved. Section 3.3 does not offer particular insight and does not blend with the previous applicative part. The paper does not offer many visualizations of the results or quantitative comparisons. These could help the reader to better understand the impact of DPO in this domain. For example, recent works suggest that DPO may be prone to overfitting [A]. I would like to know what the impact of the procedure is in the atomic structure prediction and if they observed similar problems. Similarly, although the authors offer some possible future directions, they don't really discuss failure cases of the method, and the presented limitations seem a bit generic.

[A]: A general theoretical paradigm to understand learning from human preferences, Azar et al., AISTATS 2024

**Questions:**

Adding to the points above:
1) How efficient is the proposed approach w.r.t. the competitors [9, 14, 33]?
2) From my intuition, symmetries in the atomic structures may be a relevant aspect to consider in constructing the reference dataset since it could lead to ambiguities but also provide a source of augmentation. I would like to know what is the authors' thoughts on this aspect.

**Limitations:**

The paper phrases the limitations mainly as future works but does not discuss the failure modes of the proposed approach. I suggest the authors comment on recurrent confusion cases for the observed generations.

---

> ### Author Rebuttal · Authors · 2024-08-07
>
> We thank the reviewer for the valuable comments, and answer the questions as follows.
>
> > **W1: The significance of the work.**
>
> Thank you for your kindly comment. Our paper, while applying DPO to atomic systems, introduces significant innovations tailored for this domain:
>
> - **Extension of Gaussian Paths**: We expand the typical diffusion paths from previous works[14,20] to a broader set of Gaussian paths, and ensure these paths still meet the symmetry requirements essential for accurate structural predictions, enhancing exploration of conformational space.
>
> - **Unified DPO Objective**: Our theoretical analysis ensures the universality of the DPO objective that is applicable across all Gaussian paths, not limited to the traditional diffusion paths [30].
>
> - **Automated Preference Dataset Generation**: We introduced a method to automatically generate preference datasets from the training data, reducing the need for manual annotations and lowering resource demands.
>
> - **Empirical Validation**: Our method has been validated on challenging antibody and crystal datasets, demonstrating its effectiveness and practical utility in real-world 3D structure prediction.
>
> These contributions address specific challenges in 3D structure prediction for atomic systems, offering substantial advancements over existing methods.
>
> > **W2: The general presentation could be improved.**
>
> Thank you for your constructive feedback.
>
> We acknowledge your concerns regarding the general presentation and the intergration of Section 3.3. The primary intention of our paper is to demonstrate the effectiveness of DPO across various flow models and Gaussian paths for atomic structure prediction tasks. In Section 3.3, we aimed to establish the generality of DPO by ensuring the training objective is universal for arbitrary Gaussian paths, as introduced in Section 3.1. This theoretical foundation is crucial for supporting the feasibility of the experiments detailed in Section 4, bridging the gap between theoretical derivations and practical applications.
>
> Moreover, the impact of DPO has been visualized in Figure 4 in our paper. Specifically, Figure 4 illustrates the distribution of a set of samples generated by our model, where the x-axis represents the RMSD to the ground truth and the y-axis represents the probability density. The blue curve represents the model's performance before DPO optimization, and the red curve represents performance after DPO optimization. Notably, the blue curve often exhibits a bimodal distribution, with the first peak at a lower RMSD indicating higher quality conformations, and the second peak at a higher RMSD representing conformations that deviate more from the ground truth. These higher RMSD samples often include physically implausible conformations. After applying DPO, we observe a suppression of the erroneous peak, indicating an overall improvement in the quality of the generated structures. This visualization clearly demonstrates DPO's role in enhancing model performance by effectively reducing the generation of less accurate structures. We provide additional visualizations in Figure S1 of general response, which also support the above observations.
>
> For the overfitting problem and the bad cases, one worth-noticed phenomenan is that the learning rate for DPO training affects the model performance. For instance, a high learning rate can cause the model to deviate from the initial state, and the original distribution is forgetten. We provide an example in Table S2 of the general response for the validation performance on MPTS-52 crystal structure prediction with OT-OT path as follows, where high learning rate hinders the model performance.
>
>
>
> > **Q1: How efficient is the proposed approach w.r.t. the competitors?**
>
> In assessing the efficiency of our proposed approach relative to competitors, we provide the inference time (minutes) for each method when applied to MP-20 in Table S3 of the general response. Note that DPO does not change the inference process of the flow models. Consequently, models optimized with DPO maintain comparable generation efficiency to their original versions.
>
>
> > **Q2: The role of symmetries.**
>
> Nice Question! Symmetries indeed play a crucial role in the field of atomic systems. In our study, we have carefully addressed this issue by ensuring that all proposed flow paths rigorously maintain the symmetries specified in Eq. (6) and Eq. (12). As a result, our model inherently respects these symmetries. Specifically, the predicted antibody CDRs exhibit equivariance relative to the provided context, and the modeled crystal distributions are invariant under E(3) transformations. Given this inherent compliance with symmetry requirements, we do not need to employ additional data augmentation strategies to artificially introduce symmetries.

---

> > ### Comment · Reviewer_jaPH · 2024-08-09
> > **Post-Rebuttal**
> >
> > I thank the author for their reply to my concerns and for providing further evidence on the model's learning regime comparing different learning rates, as well as a comparison of inference time. I also see that the symmetric guarantee of the paths removes the need for augmentation. However, I wonder whether the generated structures actually present such symmetries, i.e., in your experiments, have you observed whether the model has a probability of generating all the solutions within the E(3) equivalence class, or does it always generate a representative?
> >
> > About the reply to (W1), I am still a bit unsure. I understand that dataset generation and evaluation would change with a different domain/task, but the technical contributions of the work (Gaussian Paths and DPO Objective analysis) could also be applied to other applications (e.g., 3D rigid object generation, Shape completion). With this, I just want to clarify my understanding of the contribution and the possible scope of the proposed solution.
> >
> > From other reviews, I find the comments around the validation interesting. In particular:
> > - **RMSD**: this is a standard measure in the field, as it approaches 0 when the structure is perfectly aligned. However, to me, it is unclear how it considers symmetries, e.g., comparing the same structure but rotating 90° along one of the three axes could cause a large error. Also, RMSD tends to emphasize the magnitude of the errors rather than the difference in the structures.  Are these issues considered by the community? Are there other possible error measures? Maybe a discussion on these aspects could help us better understand the evaluation protocol.
> > - **Single ground truth**: Due to the problem's probabilistic nature, considering a single ground truth seems reductive. By my understanding, the intuition is that, in general, with a desirable target stable structure, the expectation is that good predictions resemble similar stability. However, if the model predicts other stable configurations that are not reflected in the ground truth, these are evaluated as wrong, so the error can be considered an upper bound rather than an exact evaluation of the method's mistakes. Is this intuition correct?
> >
> > Thank you again for your time and clarifications.

---

> > > ### Author Response · Authors · 2024-08-09
> > >
> > > Thanks for your valuable feedbacks. We would like to answer the reviewer's concerns as follows:
> > >
> > > > **Regarding Symmetry Preservation:**
> > >
> > > We appreciate the reviewer’s observation about symmetry. Indeed, our model captures the symmetric distribution. For instance, in the crystal structure prediction task, we observed that the model generates multiple structures that are equivalent under different E(3) transformations. This behavior aligns with previous works that account for such symmetries [14, 33]. We will include additional visualizations in the revised paper to further substantiate this phenomenon.
> > >
> > > > **Regarding the Scope of our Contributions:**
> > >
> > > We appreciate the reviewer's understanding of our contribution. In this work, we explore more Gaussian paths and ensure their symmetries, and generalize the DPO objective on these paths. While our current study centers on atomic systems, we agree that the framework could be extended to other 3D tasks, such as rigid object generation or shape completion, as mentioned. We believe these applications represent promising directions for future work.
> > >
> > > > **Regarding RMSD and Evaluation Metrics:**
> > >
> > > The reviewer’s concern about the symmetry-awareness of RMSD is well noted. In antibody structure prediction tasks, RMSD is calculated directly since the ground truth and generated structures share the same coordinate system, defined by the given context (e.g., antigen and framework regions). For crystal structure prediction, we employ StructureMatcher from pymatgen [22], which considers all possible alignments and returns the minimum RMSD, ensuring that transformations do not affect RMSD values.
> > >
> > > Regarding alternative metrics, in the domain of antibodies and proteins, TMScore and LDDT are also worth-mentioned. While TMScore calculates the entire large protein but not specific CDRs, and LDDT focuses on side-chain structures, RMSD remains the most suitable metric for CDR backbone generation. For crystal structure prediction, we also report match rates as a threshold-based metric to measure differences between ground truths and generated structures.
> > >
> > > > **Regarding the Single Ground Truth:**
> > >
> > > Your intuition is correct. Given the nature of data scarcity, and the fact that a low RMSD with the observed ground truth often reflects similar stability, RMSD provides a practically meaningful evaluation in the domain of atomic systems [1, 6].
> > >
> > > We sincerely appreciate your recognition of the significance of our work. Your feedback is precious for refining this paper.

---

> ### Author Response · Authors · 2024-08-12
> **Looking Forward to your Further Feedback**
>
> Dear Reviewer jaPH,
>
> Thanks for your valuable questions regarding the symmetries and evaluation methods. We hope all your concerns have been addressed, but please let us know if there is anything further you would like to discuss.
>
> Best,
>
> Authors

---

### Official Review · Reviewer_WEst · 2024-07-11

**Soundness:** 3
**Presentation:** 4
**Contribution:** 3
**Rating:** 6
**Confidence:** 3

**Summary:**

The paper introduces a framework called FlowDPO subsuming different Gaussian probability paths for flow matching models predicting the 3D structure of atomic systems. Moreover, the authors propose a method for the automatic creation of a preference pair dataset used for finetuning a pre-trained flow matching model with direct preference optimization (DPO). As examples of the structure prediction task for atomic systems, the paper explores antibody and crystal structure prediction, each with specific symmetry requirements. The choice of probability path is evaluated empirically including a validation of DPO in terms of consistent performance boosts.

**Strengths:**

* The paper is very well structured:
  - The presented formal framework abstracts the choice of different probability paths for flow matching models.
  - The general task of structure prediction of atomic systems is well motivated and explained by the two instances in form of antibody and crystal structure prediction.
  - The structure of the method section follows the training procedure: Flow Matching training, preference dataset construction, direct preference optimization.
* The paper is original: The authors apply general flow matching and direct preference optimization to the new domain of atomic system structure prediction.
* The contributions of DPO for flow matching and automatic preference dataset construction are general and therefore also applicable to other domains.
* The experimental results validate the proposed framework:
  - Exploration of different probability paths can be beneficial, as there is not always one performing the best.
  - Direct preference optimization on the proposed automatically constructed datasets consistently improves performance.

**Weaknesses:**

* Missing motivation for a generative approach to 3D structure prediction of atomic systems.
* Unclear/imprecise use of the term "hallucination". The paper claims the model can distinguish between high-fidelity and hallucinated samples (cf. 52 f) without defining what hallucinated samples really are. Recombination / generalization is usually desired for generative models.
* Self-contradictory method:
  - The framework is about flow matching models, i.e., a generative method.
  - The preference definition is solely based on distances to the ground truth training examples. This way of creating a preference dataset does not account for the possibility of multiple possible solutions given the conditioning. Optimization w.r.t. the RMSD metric is not even significantly different from the flow matching objective, begging the question for why this improves performance at all?
  - Originally, DPO is used in combination with crowd-sourced human preferences for human alignment. This type of metric is significantly different from the pixel-wise Gaussian noise and MSE regression during training of text-to-image models, for example.
* Contradictory method and evaluation:
  - The paper uses generative flow matching models.
  - The evaluation is done in terms of distances to a single ground truth, which does not account for the possibility of multiple valid solutions. However, this might be a general problem of generative approaches for regression tasks.
* The comparison with baselines is limited to generative methods.

**Questions:**

* Is there uncertainty in two given tasks, i.e., can there be multiple different valid solutions to predicting the 3D structure given the respective prior knowledge as conditioning?
  - If no, why would you use a generative approach?
  - If yes:
    - Why do you construct preferences based on distances to a single ground truth without accounting for the possibility of multiple solutions given the conditioning?
    - Why does the evaluation consider distance to a single ground truth?
* What do you understand as hallucinated samples?
* Are there non-generative (regression) baselines and if so, how is the performance compared to them?

**Limitations:**

The authors have addressed the limitations and broader impacts without any potential negative societal impacts in the appendix.

---

> ### Author Rebuttal · Authors · 2024-08-07
>
> We thank the reviewer for the valuable comments, and answer the reviewer’s questions as follows.
>
> > **W1: Missing motivation for a generative approach to 3D structure prediction of atomic systems.**
> > **Q1.1: why would you use a generative approach?**
>
>
> Thank you for your insightful question. The motivation for using generative model first arises from the scarcity of experimental data, which typically provides only a single near-stable structure. Generative models can learn and represent a distribution of potential stable structures, by maximizing the likelihood of the observed samples within the modeled distribution, filling gaps in the data. Their success in works like DiffAb, DiffCSP, DiffDock, and AlphaFold3 further supports this direction.
>
>
> > **W2: Unclear/imprecise use of the term "hallucination".**
> > **Q2: What do you understand as hallucinated samples?**
>
> Thank you for your feedback. We recognize that the term "hallucination" was used without a clear definition, leading to potential confusion. In the context of our work, "hallucination" refers to instances where the generative model produces conformations that are not only incorrect but also highly improbable from a physical view.
>
> As depicted in Figure 4, the generative model's output distributions are represented by two curves on where the x-axis measures the RMSD from the ground truth and the y-axis indicates the probability. The blue curve, which illustrates the model's performance prior to the application of DPO, often displays a bimodal distribution. The first peak, with lower RMSD, indicates higher quality predictions, while the second peak, with higher RMSD, corresponds to conformations that deviate significantly from the ground truth.
>
> Upon closer examination, we found that this second peak often includes physically implausible conformations. In scenarios without DPO, the probability associated with this erroneous peak can surpass that of the more accurate peak. We have termed this phenomenon as "hallucination," indicating the unexpected high probability of low-quality, implausible generations, which DPO effectively helps to suppress.
>
> If the term "hallucination" leads to ambiguity, we are open to describing it more directly as incorrect or low-quality generations.
>
> > **W3: Self-contradictory method.**
> > **Q1.2: Why do you construct preferences based on distances to a single ground truth without accounting for the possibility of multiple solutions given the conditioning?**
>
>
> Thank you for your insightful comments. As mentioned above, high RMSD values often indicate low-quality conformations. Our DPO approach treats low RMSD structures as preferred, guiding the model to reduce low-quality generations. This adapts DPO to a quantitative metric relevant to 3D atomic structures, ensuring objectivity and scalability without relying on costly expert evaluations.
>
>
> > **W4: Contradictory method and evaluation.**
> > **Q1.3: Why does the evaluation consider distance to a single ground truth?**
>
> Thank you for your observations. We employ RMSD as an evaluation metric, recognizing its widespread acceptance and proven utility in our domain. Experimentally measured structures, which serve as benchmarks in our evaluations, are generally near a stable state. Consequently, structures generated close to these experimental conformations are typically more reliable and indicative of stability. Moreover, this metric has been widely adopted in the field, including in seminal works such as AlphaFold3, and is well-recognized within our community.
>
> > **W5: The comparison with baselines is limited to generative methods.**
> > **Q3: Are there non-generative (regression) baselines and if so, how is the performance compared to them?**
>
> Thanks for your advice. We utilize the same backbone model (MEAN) directly for a regression task as an additional baseline. The results are shown in Table S1 in the general response. We observe that the generative models surpass the regressive model on 4 of the 6 CDRs, notably in the most variable and critical regions, CDR-H3 and CDR-L3. Additionally, we report not only the mean RMSD across 20 generations for each generative model but also the minimum RMSD. It can be seen that the minimum RMSDs are significantly lower, showcasing that the generative models not only provide predictions that are closer to the observed reference structure but also demonstrate the ability to generate multiple viable structures around the stable state.
>
>
> > **Q1: Is there uncertainty in two given tasks, i.e., can there be multiple different valid solutions to predicting the 3D structure given the respective prior knowledge as conditioning?**
>
>
> Certainly, there is inherent uncertainty in predicting the 3D structures of molecules due to the dynamic nature of molecular conformations. These conformations are not static or fixed; instead, there are many possible structures clustering around the energy minima[a]. Therefore, a generative modeling approach is employed to capture the probability distribution of these stable structures, aiming to maximize the likelihood of observing conformations close to these energy minima. Regarding the use of RMSD as a metric, it serves to assess how closely the generated structures align with a known stable conformation. Since the plausible structures only deviate slightly from each other, structures distinct from the observed stable conformations are still less reliable. Therefore, although this approach does not account for the multiplicity of valid solutions, it provides a practical measure of deviation from a recognized ground truth, facilitating the evaluation of model performance in generating physically plausible structures.
>
> [a] Fernández-Quintero, Monica L., et al. "CDR-H3 loop ensemble in solution–conformational selection upon antibody binding." MAbs. Vol. 11. No. 6. Taylor & Francis, 2019.

---

> > ### Comment · Reviewer_WEst · 2024-08-09
> > **Post-Rebuttal**
> >
> > I thank the authors for their clarifications regarding my concerns and lack of domain knowledge. Especially, the last answer to my Q1 was helpful for my general understanding and the motivation of using a generative approach.
> >
> > - *Motivation*: My understanding is that flow matching is used to generate plausible structures close to known stable conformations. Given domain knowledge, predicted structures further away from known ones w.r.t. RMSD as metric are considered to be incorrect / low-quality (hallucinations in the paper) and are therefore suppressed by DPO. Could you confirm, whether my understanding is correct?
> > - W3: Could you elaborate on the difference between the RMSD metric and the flow matching objective (see W3.2)? Is the training loss, which is essentially also a squared distance between the model prediction and the ground truth vector field towards a known stable conformation, not very similar to the RMSD metric?
> >   - Could the reason for undesired hallucinations be that the assumption of Gaussian paths is not well-aligned with the distance metric relevant in this domain?
> >   - In other domains, there have been also approaches to "mix in" additional losses (e.g. to improve quality of novel views for 3D object generation). Would that also be an option (alternative to DPO) here, i.e., obtain a single-step estimation for $x_0$ using the predicted vector field, and add a RMSD loss w.r.t. the ground truth structure?
> > - W5: Thank you for providing a regression-based baseline. I see the benefits of a generative approach for sampling multiple candidates and possibly choosing ones closer to known structures. However, for CDR-L1, the regression baseline performs significantly better, which is surprising to me. Could you please clarify this?
> >
> > Thank you for your time and efforts.

---

> ### Author Response · Authors · 2024-08-10
>
> We thank the reviewer for the constructive comments and address the follow-up questions as follows:
>
> > **Motivation:**
>
> Yes, your understanding is correct. We utilize flow matching models to generate candidate structures, and those with high RMSDs are considered low-quality. These low-quality predictions, referred to as "hallucinations" in the paper, are suppressed using the DPO proposed in Section 3.3.
>
> > **Regarding W3:**
>
> There is indeed a similarity between the flow matching training objective and the RMSD metric. In fact, the rectification strategy suggested by the reviewer—to align the model-predicted $x_0$ at each step with the ground truth—is essentially equivalent to the original flow matching objective. Taking the OT path as an example, the flow matching objective on the vector field is to minimize $\\\|v_t-\frac{x_t-x_0}{t}\\\|_2^2$, which can be reparameterized as $\\\|x_0-(x_t-v_t\cdot t)\\\|_2^2$.
>
> Then why does DPO outperform the original model? The key difference lies in the training objective. Unlike the original loss, which solely fits predictions to the ground truth, DPO not only guides the model towards the "win" cases but also corrects inaccuracies in the "lose" cases, steering the model away from low-quality predictions.
>
> Moreover, regarding the potential causes of hallucinations, one possible factor is the imperfect modeling of intermediate states. In a multi-step generation process, small errors can accumulate, leading to inaccuracies in the final structure. The DPO objective mitigates this by correcting the outputs from the inaccurate ("lose") cases at each timestep, thereby improving overall performance.
>
> > **Regarding W5:**
>
> The performance of the regression baseline on CDR-L1 can be attributed to the relatively low diversity of this particular CDR, where the structure patterns are similar. Hence the regression task might be better in this specific case. However, research has primarily focused on more flexible regions like CDR-H3 and CDR-L3, where structure complexity and context-dependency make prediction tasks more challenging. In these cases, generative models exhibit significantly better performance.
>
> We appreciate the kindly discussions with the reviewer, and look forward to your further feedback.

---

> ### Comment · Reviewer_WEst · 2024-08-12
>
> The rebuttal addresses most of the mentioned weaknesses and questions. After carefully considering all reviews and answers by the authors, I am still advocating for accepting the paper by increasing my rating to: 6 Weak Accept (cf. edited rating)

---

> > ### Author Response · Authors · 2024-08-12
> > **Thank you**
> >
> > Dear Reviewer WEst,
> >
> > Your valuable comments do help improve this paper. Thank you very much!
> >
> > Best,
> >
> > Authors

---

### Official Review · Reviewer_SYsR · 2024-07-12

**Soundness:** 3
**Presentation:** 2
**Contribution:** 2
**Rating:** 5
**Confidence:** 3

**Summary:**

This paper introduces FlowDPO, a framework that predicts 3D structures of atomic systems using diffusion flow matching models. To suppress hallucinations and improve sample quality, Direct Preference Optimization (DPO) is adopted to finetune a pretrained model using a preference dataset consisting of winning and losing pairs, which are selected based on their distance to the ground truth. Details on the flow matching model and DPO training paradigm has been discussed. Experiments have been conducted on antibody and crystal structure prediction tasks, demonstrating better accuracy than the two baselines (P-cG-SchNet and CDVAE).

**Strengths:**

- This work introduces the application of diffusion models with DPO into the field of atomic structure predictions. Unlike the original diffusion DPO paper [30], where samples have to be ranked by humans, this task benefits from known ground truths from a training dataset, allowing the winning and losing pairs to be automatically constructed by comparing the distances to the ground truth.
- Experimental results demonstrate the advantage of the proposed method over other generative models, and ablation studies have revealed the contribution of the DPO fine-tuning.

**Weaknesses:**

- The motivation for symmetry preservation and transform invariance of the flow trajectory is unclear and seems not quite relevant to the main idea of using DPO to improve sample accuracy. The theoretical justification for the trajectory needing to be transform invariant is not well explained, apart from the empirical results.
- In L57, the authors claim that they are "the first to theoretically prove the compatibility of DPO with arbitrary Gaussian paths by deriving a unified objective." However, I can hardly find anything relevant to this claim, and the paper seems more focused on application rather than theory.
- In general, the contribution is limited, considering that diffusion DPO is a known method in [30], which is theoretically universally applicable to all diffusion flow matching models. While this work touches on the new field of atomic system prediction, using a universally applicable method in this field does not seem very novel to me.
- The writing of this paper has some issues, making the presentation a bit unclear:
  - The overall structure is not focused. While the main idea of this work is the application of diffusion DPO rather than proposing a new theory, the paper spends a lot of space on preliminaries with very specific examples. The main body, especially the experiments section, is short and lacking in detail.
  -  The subscript 'i' is a bit confusing as it sometimes denotes the index of a data sample and sometimes the index of a timestep.

**Questions:**

As mentioned in the weaknesses, I do not understand the motivation for symmetry preservation and transform invariance, and the claim about the theoretical proof.

**Limitations:**

Yes

---

> ### Author Rebuttal · Authors · 2024-08-07
>
> Thanks for your constructive comments! We provide more explanations to address your concerns as follows.
>
> > **W1: The motivation for symmetry preservation and transform invariance of the flow trajectory is unclear and seems not quite relevant to the main idea of using DPO to improve sample accuracy. The theoretical justification for the trajectory needing to be transform invariant is not well explained, apart from the empirical results.**
>
> Thanks for your feedback. In many atomic systems, symmetry indeed plays a fundamental role, as the physical laws governing these systems are invariant under certain transformations. More specifically, for antibody structure prediction, the E(3) equivariance of the trajectory in Eq. (7) ensures that the predicted CDRs are equivariant to the given context [20]. Similarly for crystal structure prediction, the periodic E(3) equivariance described in Eq. (13-14) ensures that the predictions retain the inherent symmetries in crystal structures [14]. As we also explore various new flow paths beyond those in previous studies [14,20], it is imperative to ensure that these newly proposed paths still adhere to the necessary symmetries, thereby validating their correctness. We will expand on these points in the revised paper to more clearly articulate the role and necessity of maintaining these symmetries.
>
> > **W2: In L57, the authors claim that they are "the first to theoretically prove the compatibility of DPO with arbitrary Gaussian paths by deriving a unified objective." However, I can hardly find anything relevant to this claim, and the paper seems more focused on application rather than theory.**
>
> Thank you for your observation. We apologize for any ambiguity in our presentation. In Section 3.3, we generalize the feasibility of DPO from diffusion-based VP paths to arbitrary Gaussian paths. Specifically, as detailed in Eq. (27), we demonstrate that Gaussian paths can be universally discretized using conditional mean values parameterized by $x_i$ and $x_0$. As $x_i$ is given, the KL divergence term in Eq. (26) can be reparameterized by the MSE of the ground truth $x_0$ and $x_{0,\theta}$ predicted by the flow model. This formulation allows us to derive a unified DPO training objective applicable to all Gaussian paths, as outlined in Eq. (28). We acknowledge the need for clearer exposition on this theoretical contribution and will enhance the relevant sections in our paper to better articulate these derivations and their implications.
>
> > **W3: In general, the contribution is limited, considering that diffusion DPO is a known method in [30], which is theoretically universally applicable to all diffusion flow matching models. While this work touches on the new field of atomic system prediction, using a universally applicable method in this field does not seem very novel to me.**
>
> Thanks for pointing this out. As mentioned above, while it is true that Diffusion-DPO [30] establishes the framework of DPO within the context of diffusion models like DDPM, which predominantly utilize the Variance Preserving (VP) path, our contribution extends this framework to arbitrary Gaussian paths and adapts to a broader class of flow models, and demonstrates the practical applicability on 3D structure prediction tasks.
>
> > **W4: The overall structure is not focused. While the main idea of this work is the application of diffusion DPO rather than proposing a new theory, the paper spends a lot of space on preliminaries with very specific examples. The main body, especially the experiments section, is short and lacking in detail.**
>
> Thanks for your feedback. We acknowledge that the current structure of the paper may not optimally highlight the core contributions of our work. Our primary aim is to demonstrate the efficacy of DPO across a range of flow models characterized by arbitrary Gaussian paths, specifically within the context of 3D structure prediction tasks.
>
> To establish this, we introduced multiple flow paths that extend beyond the typical diffusion-based paths previously explored in literature [14,20]. These paths are crucial for demonstrating the broad applicability of DPO in handling diverse flow models. Additionally, maintaining the necessary symmetries for each specific prediction task is essential for the design of each proposed path, which we underscore in Section 3.1. We derives the generality of DPO in Section 3.3, and demonstrate its effectiveness for multiple tasks and multiple flow paths in the experiments. We will carefully reorganize the content to ensure a more logical flow that aligns with the main contributions of our work.
>
> > **W5: The subscript 'i' is a bit confusing as it sometimes denotes the index of a data sample and sometimes the index of a timestep.**
>
> Thanks for your kind advice. We would like to change the notation of timesteps into 's' to avoid confusion.
>
> Thank you again for your valuable suggestions. We believe we have addressed your concerns and kindly hope for your feedback and reconsideration of the scores.

---

> > ### Comment · Reviewer_SYsR · 2024-08-14
> >
> > Thank you for the detailed feedback. After reading the other reviewers' comments, I agree that the originality of the paper in applying DPO to the field of atomic structure prediction is a valuable contribution.
> >
> > My remaining concerns are still about the theoretical aspects. While the transform invariance of the ground truth flow trajectory can be guaranteed (though I still do not see a theoretical proof of why certain paths lead to certain invariances and not others), I believe the main point is missed: the flow prediction network should be designed to be transform invariant, which would make it a better fit to the ground truth trajectory. The experiments on different paths provide interesting empirical insights but do not seem relevant to the transform invariance properties, in my opinion.
> >
> > To summarize, I am inclined to raise my rating towards acceptance, but I strongly recommend that this paper place more emphasis on the application and DPO aspects rather than on trajectory invariance.

---

> > > ### Author Response · Authors · 2024-08-14
> > >
> > > We appreciate the reviewer's constructive suggestions. We are committed to reorganizing the paper by incorporating more analyses into the main text, including additional visualizations and explanations, to better highlight the effectiveness of DPO in reducing low-quality generations. To focus more on DPO and its impact, we are also willing to move the specific details of flow paths to the appendix, where we will additionally provide explanations on the necessity of symmetries and visualizations to validate the preservation of these symmetries (suggested by reviewer jaPH).
> > >
> > > Additionally, we would provide theoretical proofs on the symmetries. While the detailed proofs depend on specifc paths, the key insight is that an invariant or equivariant final distribution can be derived from an invariant or equivariant prior, and an equivariant transition process [34]. The former is ensured by the isotropy of the Gaussian distribution, and the latter is achieved through the design of backbone models. We will detail the design of these models for each task and provide proofs of their equivariance.
> > >
> > > Thank you again for your valuable advice in enhancing the clarity and focus of this paper.
> > >
> > > [34] Xu, Minkai, et al. "GeoDiff: A Geometric Diffusion Model for Molecular Conformation Generation." International Conference on Learning Representations.

---

> > > ### Author Response · Authors · 2024-08-14
> > >
> > > Dear Reviewer SYsR,
> > >
> > > Thanks again for your suggestions. We will definitely revise our paper accordingly. As you mentioned considering raising the rating, we would appreciate knowing if your concerns have now been fully addressed.
> > >
> > > Best,
> > >
> > > Authors

---

> ### Author Response · Authors · 2024-08-12
> **Looking Forward to your Feedback**
>
> Dear Reviewer SYsR,
>
> Thanks again for your insightful comments. We have addressed your concerns by clarifying the motivations and overall structure of this work in our rebuttal. Please let us know if you have any further questions.
>
> Best regards,
>
> Authors

---

### Author Rebuttal · Authors · 2024-08-07

We sincerely thank all reviewers and ACs for their time and efforts on reviewing the paper. We are glad that the reviewers recognized the contributions of our paper, and appreciate the reviewers for their insightful comments. We provide additional visualizations and experment results in the supplementary PDF file for more details. We summarize the extra contents as follows.

- **Table S1** combines the results with regressive baselines, highlighting the superiority of generative models in predicting flexible CDRs.
- **Table S2** explores the impact of overfitting by comparing the effects of different learning rates.
- **Table S3** evaluates the efficiency of various models.
- **Figure S1** offers additional visualizations of crystals, illustrating the impact of DPO by reducing low-quality generations.

---

### Decision · Program_Chairs · 2024-09-25

**Decision:**

Accept (poster)

**Comment:**

Initially, this paper got borderline scores also due to slightly out-of-domain reviewers that are familiar with diffusion and flow matching but not with the task of atomic system structure prediction, which is tackled here. Initially, reviewers were questioning the use of generative models in the given setting, the significance of contribution, i.e. if the generalized DPO actually has relevant impact on the results of the given task, and the need for the theoretical contributions.

The discussion phase of this paper was very valuable, helped to improve the paper and clarified the intuition and details behind the presented approach. After the discussion phase, the reviewers unanimously converged to accept scores, being convinced by the detailed answers of the authors. Without being an in-domain expert either, I agree with the reviewers. I think the paper is interesting and of high quality, showcasing successful application of diffusion flow matching in 3D structure prediction tasks. I follow with an accept recommendation.